# Data Compression Based on Stacked RBM-AE Model for Wireless Sensor Networks

**DOI:** 10.3390/s18124273

**Published:** 2018-12-04

**Authors:** Jianlin Liu, Fenxiong Chen, Dianhong Wang

**Affiliations:** School of Mechanical Engineering and Electronic Information, China University of Geosciences, Wuhan 430074, China; liujianlin@cug.edu.cn (J.L.); wangdh@cug.edu.cn (D.W.)

**Keywords:** wireless sensor networks, data compression, stacked RBM, transfer learning, energy consumption optimization

## Abstract

Data compression is very important in wireless sensor networks (WSNs) with the limited energy of sensor nodes. Data communication results in energy consumption most of the time; the lifetime of sensor nodes is usually prolonged by reducing data transmission and reception. In this paper, we propose a new Stacked RBM Auto-Encoder (Stacked RBM-AE) model to compress sensing data, which is composed of a encode layer and a decode layer. In the encode layer, the sensing data is compressed; and in the decode layer, the sensing data is reconstructed. The encode layer and the decode layer are composed of four standard Restricted Boltzmann Machines (RBMs). We also provide an energy optimization method that can further reduce the energy consumption of the model storage and calculation by pruning the parameters of the model. We test the performance of the model by using the environment data collected by Intel Lab. When the compression ratio of the model is 10, the average Percentage RMS Difference value is 10.04%, and the average temperature reconstruction error value is 0.2815 °C. The node communication energy consumption in WSNs can be reduced by 90%. Compared with the traditional method, the proposed model has better compression efficiency and reconstruction accuracy under the same compression ratio. Our experiment results show that the new neural network model can not only apply to data compression for WSNs, but also have high compression efficiency and good transfer learning ability.

## 1. Introduction

Research has found sensor networks suitable to collect environment information, such as temperature, light and humidity in harder-to-reach areas. In recent years, with the development of information technology, wireless sensor networks (WSNs) have played an important role in a wide range of disciplines [1]. Unmanned vehicles need GPS and accelerometer to locate themselves, and to get information regarding their environments using a camera and lidar [2]. By fusing these multi-modal sensor data, they can also predict the moving trajectory of nearby objects. Similarly, environmental monitoring applications need temperature, humidity, wind direction, etc. The increase in WSNs applications and sensor nodes have led to an explosion in the amount of sensing data. If the sensing data collected by WSNs are directly sent to a gateway, it will not only consume a lot of power but also increase the probability of transmission errors. Because a sensor node’s storage and portable energy resources is limited, it is important to design an energy-saving and redundant data compression scheme for WSNs. These factors have motivated efficient technologies to reduce energy consumption and extend the lifetime of WSNs.

WSNs’ data collection methods include direct transmission to the base station, multi-hop forwarding, data aggregation, and modeling with coding. Data aggregation is an important technology for data processing in WSNs. By aggregating the collected or received data, the amount of data to be transmitted can be effectively reduced [3]. In the existing data aggregation algorithm research, the node usually compares the relationship between its perceptual data and the surrounding node data. If the data is found to be close, it will not be submitted in order to reduce the transmission of redundant data. Although these aggregation algorithms reduce the amount of data and save the energy consumption of the node, they cause a loss of node data. In addition, when aggregating, the outlier detection is expensive and could introduce delays [4]. The Slepian-Wolf coding applies distributed source coding technology to perform non-collaborative data compression at the sources. However, it is not practical due to the lack of prior knowledge of the data correlation structure [5]. Model-based compression algorithms such as APCA [6] and PWLH [7] also have good compression ratio. However, they result in loss of sensing data, as they approximate data with temporal and spatial locality. There is research that shows that modeling with the encoding method is better for data transmission of WSNs [8]. It first models the original sensing data of the sensor node, following which the modeled weights are sent to the base station, which uses the weights to compress the original sensing data.

Compressed sensing (CS) [9] provides a new data compression method for WSNs. The basic idea is that if the signal is sparse or compressible at a certain level, it can be reconstructed from a small number of linear measurements lower than the Shannon–Nyquist limit [10]. In recent years, the study of CS has led to many significant developments in the field of signal processing, including novel sub-Nyquist sampling strategies and a veritable explosion of work in sparse approximation and representation [11,12]. In Ref. [13], the CS technology proposes a method to reduce the sensing data traffic for WSNs without the need for adapting to the data correlation structure. In Ref. [14], the CS principle was used as a compression and forwarding scheme to minimize the transmission data. At present, the CS requires a large amount of memory space to store the random sampling operator when the signal amplitude is large. Therefore, the classic CS is not directly applied to largescale application, and it will inevitably increase the computational complexity of encoding, so the CS compression is limited by the mobile embedded processor. At the same time, it is difficult to apply largescale WSNs while its energy is constrained. The analog CS technique is a novel strategy to sample and process sparse signals at a sub-Nyquist rate [15]. In Ref. [16], the problem of energy consumption for sensor nodes performing CS and DCS was addressed when both digital and analog CS were considered. The minimum energy CS-based data aggregation problem was studied to minimize the total energy consumption of a WSN in collecting sensing data from the whole network [17]. In Ref. [18], the CS based signal and data acquisition for WSNs was proposed, and a cluster-sparse reconstruction algorithm was proposed for in-network compression to achieve accurate signal recovery and energy efficiency. Although the segmented linear compression method can be used to reduce the dimension of high compression ratio by using polynomial approximation in the form of segmentation, it has poor smoothness, poor precision and abnormal change. In Refs. [19,20], the data compression algorithms based on spatiotemporal correlation for WSNs were summarized. In Ref. [21], the run length coding method was proposed, which is suitable to compress the sensing data for WSNs. A lightweight data compression algorithm based on spatial correlation was proposed in ICACT [22], which saves more energy than wavelet compression in the case of the same distortion rate. However, it is mentioned above that the learning ability of the shallow learning algorithm is very weak for advanced features.

The deep learning method can extract detection data in multi-level features, which makes the system have a strong data fitting ability by learning the deep features in the data. At present, there are few researches on how to use the deep learning model to compress the sensing data for WSNs. In Ref. [23], a method of using principal components analysis (PCA) to reduce the dimensionality of data was proposed. In Ref. [24], the unification of machine learning approaches with CS was also addressed, in which feed-forward deep neural network structures are used to aid CS signal reconstruction. In Ref. [25], a data compression algorithm based on the stacked Auto-Encoder (SAE) was proposed. This method combines the SAE and the cluster routing protocol. Compared with the traditional compression algorithm, the SAE can improve the accuracy of data fusion by 7.5%. In Ref. [26], a deep convolution network for ECG signal compression is proposed, which needs a large amount of calculation. The Restricted Boltzmann Machine (RBM) is a probabilistic model for a density over observed variables that uses a set of hidden variables (representing presence of features). In the standard RBM, all observed variables are related to all hidden variables by different parameters. It was widely used in classification and generation. In Ref. [27], the Convolutional RBM (C-RBM) was developed to achieve object detection. The C-RBM is a variant of the RBM model in which weights are shared to respect the spatial structure of images. In Ref. [28], a pre-training algorithm for the Deep Boltzmann Machines (DBMs) was described. In Ref. [29], a CS reconstruction algorithm was proposed, which amounts to two nested inference problems, one on the CS observation-matching problem, and the other on the RBM model. Compared with variational autoencoders (VAE) and generative adversarial networks (GAN), RBM has the simplest network structure and the minimum number of parameters. Correspondingly, RBM has a small computational energy consumption and is more suitable for use in WSNs with limited energy. In our survey, there is currently no research on how to directly use RBM to compress the sensing data for WSNs. Currently, we mainly explore how to use RBM to compress the sensing data. In our next research, we will also explore how to use VAE, GAN and other deep learning models to compress sensing data.

In this work, we combined the RBM generation model with the nonlinear learning method of deep learning theory, and propose the Stacked RBM-AE model, which can compress and reconstruct the sensing data for WSNs. We use four standard RBMs of different sizes to form the encoder and decoder of the Stacked RBM-AE model. The Stacked RBM-AE model compresses the sensing data by using its mathematical characteristics. In our experiments, we explore the performance of the model and compare it with other compression methods, and we propose an energy optimization method of the model to reduce the energy consumption of calculation. The contributions of this paper are summarized as follows:We developed a hybrid model named Stacked RBM-AE that combines four standard RBMs with the feature of auto-encoder to compress and reconstruct the sensing data.We proposed a new method of data compression, which has better reconstruction accuracy than the traditional algorithm under the same compression ratio.We proposed an energy optimization method by pruning the model parameters, and we analyzed the efficiency of pruning different proportion model parameters on the reconstruction accuracy of the Stacked RBM-AE model under the same compression ratio.

The remainder of this paper is organized as follows: Section 2 introduces the architecture of the Stacked RBM-AE, and the details of Stacked RBM-AE training. Section 3 discusses the results of the experiments. Specially, we conduct several tricks on energy optimization. In Section 4, we sum up our algorithm, and discus further work.

The code of the Stacked RBM-AE model is available online (https://github.com/LJianlin/Stacked-RBM-AE).

## 2. Stacked RBM-AE Model

### 2.1. Architecture of the Stacked RBM-AE

The Stacked RBM-AE model contains two parts. The first part of the Stacked RBM-AE model we call encoder E, which includes four standard RBMs (in Figure 1 and Figure 2).

Each standard RBM is an undirected graph model, which includes two layers. The vector $v\;\left(v_{1},v_{2},\ldots ,v_{n}\right)$ represents the visual layer, which is the input vector with *n* scalar components from the training dataset; The vector $h\;\left(h_{1},h_{2},\ldots ,h_{m}\right)$ represents the hidden layer vector with m scalar components. The standard RBM takes the state space $\left(v,h\right)\in {\left\{0,1\right\}}^{m+n}$. The goal of RBM is to find $P_{data}\left(v\right)$ the unknown true high dimensional distribution of the visual layer variables [30]. To achieve this goal, the high-dimensional distribution of the training dataset is modeled to get P(v|θ), where the sample distribution depends on the model parameters θ [31].

For the second part of the Stacked RBM-AE model, firstly we use the characteristics of auto-encoder to flip the encoder E to get a symmetrical scale decoding output named decoder D; the structure is shown in Figure 3. The initial input of the encoder E is used as the input of the multilayer auto-encoder. Our goal is to satisfy the top output approximately equal to the bottom input.

### 2.2. Details of Stacked RBM-AE Training

#### 2.2.1. Pre-training of standard RBM

For the standard RBM, v denotes the activation probability of all visual units, and h denotes the activation probability of all hidden layer units. The model parameter θ contains three parameters: W, b and c. v and h are the binary column vectors of 1×n and 1×m, respectively; W is the real-valued matrix of n×m, which represents the weights of the units connected; b is the bias of visual layer units in n×1 size; c is the bias of the hidden layer units in m×1 size. We use $W_{ij}$ to represent the weights between $v_{i}$ and $h_{j}$. $b_{i}$ represents the bias of $v_{i}$. $c_{j}$ represents the bias of $h_{j}$.

In the pre-training phase, we use CD algorithm [32] to train four standard RBMs of different sizes in turn. Algorithm 1 shows the procedure of the pre-training of standard RBM. It first calculates the activation probability of all hidden layer units and the activation probability of all visual layer units, and then updates model parameters by minimizing the model loss.


**Algorithm 1: Pre-training of standard RBM**
**Input:** A mini-batch of training data set **S**, the number of train iteration *iter*, the learning rate *α***Initialization:**W,b,c = 0 **While**
*i < iter*
**do**  The hidden layer units activation probability h=sigmoid(c+SW)  Get h(1) by one step Gibbs sampling for h  The visible layer units activation probability v=sigmoid(b+Wh(1))  Update parameters W,b,c according to (7)–(9)  *i + +***end**

For the standard RBM, the activation state of each hidden layer unit is conditionally independent for a given state of the visible layer unit. So, we can get the activation probability of $h_{j}$:(1)$$P\left(h_{j}=1|v\right)=sigmoid\left(c_{j}+\underset{i}{\sum }v_{i}W_{ij}\right)$$

Similarly, when a state of a hidden layer unit is given, the activation probability of the visual layer unit is also conditionally independent:(2)$$P\left(v_{i}=1|h\right)=sigmoid\left(b_{i}+\underset{j}{\sum }W_{ij}h_{j}\right)$$

We define the energy function as: (3)$$E\left(v,h|\theta \right)=-\overset{n}{\underset{i=1}{\sum }}\overset{m}{\underset{j=1}{\sum }}v_{i}W_{ij}h_{j}-\overset{n}{\underset{i=1}{\sum }}v_{i}b_{i}-\overset{m}{\underset{j=1}{\sum }}h_{j}c_{j}$$

The energy function means that there is an energy value between each visual layer unit and each hidden layer unit. By indexing and regularizing the energy function, we get the joint probability of v and h:(4)$$P\left(v,h|\theta \right)=\frac{\exp\left(-E\left(v,h|\theta \right)\right)}{\sum_{v,h}\exp\left(-E\left(v,h|\theta \right)\right)}\;=\frac{\exp\left(\sum_{i=1}^{n}\sum_{j=1}^{m}v_{i}W_{ij}h_{j}+\sum_{i=1}^{n}v_{i}b_{i}+\sum_{j=1}^{m}h_{j}c_{j}\right)}{\sum_{v,h}\exp\left(\sum_{i=1}^{n}\sum_{j=1}^{m}v_{i}W_{ij}h_{j}+\sum_{i=1}^{n}v_{i}b_{i}+\sum_{j=1}^{m}h_{j}c_{j}\right)}$$

We use the likelihood function derivation method to get the solution of parameters. The joint probability distribution P(v,h|θ) is known. By summing all the states of the hidden layer unit, the edge distribution P(v|θ) of the visible layer unit set can be obtained:(5)$$P\left(v|\theta \right)=\frac{\sum_{h}\exp\left(-E\left(v,h|\theta \right)\right)}{\sum_{v,h}\exp\left(-E\left(v,h|\theta \right)\right)}\;=\frac{\sum_{h}\exp\left(\sum_{i=1}^{n}\sum_{j=1}^{m}v_{i}W_{ij}h_{j}+\sum_{i=1}^{n}v_{i}b_{i}+\sum_{j=1}^{m}h_{j}c_{j}\right)}{\sum_{v,h}\exp\left(\sum_{i=1}^{n}\sum_{j=1}^{m}v_{i}W_{ij}h_{j}+\sum_{i=1}^{n}v_{i}b_{i}+\sum_{j=1}^{m}h_{j}c_{j}\right)}$$

The representation based on the energy function is simply an alternative to the product representation of the factor [33]. According to Equation (5), we can get θ by maximizing P(v|θ).

As ln(x) is increnebted, $\arg\;\underset{\theta }{\max}\;P\left(v|\theta \right)$ and $\arg\;\underset{\theta }{\max}\ln\left(P\left(v|\theta \right)\right)$ are equivalent. Thus, we define the train goal function L(θ) as:(6)$$L\left(\theta \right)=-\ln\prod_{t=1}^{T}P\left(v^{\left(t\right)}|\theta \right)=-\sum_{t=1}^{T}\ln P\left(v^{\left(t\right)}|\theta \right)$$

T is the number of train set. $\arg\;\underset{\theta }{\min}L\left(\theta \right)$ uses the gradient descent method [34]. In the direction of $\frac{\partial L\left(\theta \right)}{\partial \left(\theta \right)}$, L(θ) becomes the fastest, we can find the minimum value of function by finding an optimal parameter adjustment stride in this direction. 

We use α to represent the learning rate of the algorithm, and the parameters update formula can be defined as: (7)$$W_{ij}=W_{ij}+\alpha \left[P\left(h_{j}=1|v\right)v_{i}-P\left(h_{j}=1|v^{\left(k\right)}\right)v_{i}^{\left(k\right)}\right]$$
(8)$$b_{i}=b_{i}+\alpha \left(v_{i}-v_{i}^{\left(k\right)}\right)$$
(9)$$c_{j}=c_{j}+\alpha \left[P\left(h_{j}=1|v\right)-P\left(h_{j}=1|v^{\left(k\right)}\right)\right]$$

$v^{\left(k\right)}$ is the state of v after ***k*** step Gibbs sampling.

#### 2.2.2. Retraining of Stacked RBM-AE

We initialize the Stacked RBM-AE model by using pre-trained standard RBM parameters, but the pre-trained parameters are only applicable to standard RBM, and we do not achieve good performance on the Stacked RBM-AE model. Therefore, we use the pre-trained parameters to retrain the Stacked RBM-AE model and use the BP algorithm [35] to fine tune the model parameters. Algorithm 2 shows the procedure of the retraining of Stacked RBM-AE. It first calculates the encoder output e and the decoder output, and then calculates the model loss according to Equation (13), finally updating model parameters. The encoder output and the decoder output are calculated by the Stacked RBM.


**Algorithm 2: Retraining of Stacked RBM-AE**
**Input:** A mini-batch of training data set **S**, the number of train iterations *iter*, the learning rate *α*, the regularization parameter *λ***Initialization:**W,b,c from the pre-trained standard RBM parameters**While**
*i < iter*
**do**  The encoder output e according to (10)  Get e(1) by one step Gibbs sampling for e  The decoder output d according to (11)  The model loss **L** according to (13)  Update parameters W,b,c according to (7)–(9)  *i + +***end**

The encoder output of the Stacked RBM-AE model is calculated by four different sizes of RBMs in turn, as shown in Figure 2. We use $W_{i}$, $c_{i}$ and $b_{i}$ to represent the parameters of the i layer standard RBM in the model. The encoder output is calculated by:(10)$$E\left(h\right)=\prod_{i=1}^{4}sigmoid\left(b_{i}+\sum W_{i}h_{i}\right)$$
where $h_{i}$ is the activation probability of all hidden layer units in the $i_{th}$ RBM. Similarly, the decoder output is calculated by:(11)$$E\left(v\right)=\prod_{i=1}^{4}sigmoid\left(c_{i}+\sum v_{i}W_{i}\right)$$
where $v_{i}$ is the activation probability of all visual layer units in the $i_{th}$ RBM. For the input data normalized to [0,1], they will be given the probability meaning that each neural unit value of the Stacked RBM-AE model still obeys the Bernoulli distribution of {0,1}. The normalized real value of the operation is considered to be the probability that the current unit has a value of 1. Then for a single training sample, the objective or cost function that minimizes the cross entropy is:(12)$$loss\left(f\left(v^{\left(t\right)}\right)\right)=-\sum_{j=1}^{m}\left(v_{j}^{\left(t\right)}\log\left(v_{j}^{\left(t\right)re}\right)+\left(1-v_{j}^{\left(t\right)}\right)\log\left(1-v_{j}^{\left(t\right)re}\right)\right)$$

For the training dataset, the loss of the Stacked RBM-AE model is defined as:(13)$$loss\left(\theta \right)=-\frac{1}{T}\left[\sum_{t=1}^{T}\left(\sum_{j=1}^{m}\left(v_{j}^{\left(t\right)}\log\left(v_{j}^{\left(t\right)re}\right)+\left(1-v_{j}^{\left(t\right)}\right)\log\left(1-v_{j}^{\left(t\right)re}\right)\right)\right)\right]\;+\frac{\lambda }{2}\sum_{l=1}^{lay-1}\sum_{tl=1}^{s_{l}}\sum_{tla=1}^{s_{l+1}}{\left(W_{\left(tla\right)\left(tl\right)}^{l}\right)}^{2}$$
where l is layer number. sl represents the unit numbers of $l_{th}$ layer, tl and tla represents the current layer number and the next layer number. The second term is the weights penalty item, which is used to prevent over-fitting. *λ* is the regularization parameter in the penalty item. 

## 3. Results and Discussion

### 3.1. Preprocessing Dataset

The experimental data was collected by the Intel Lab wireless sensor network research team from the University of California, who placed 54 sensors to collect environmental data information in the laboratory from 28 February to 5 April 2004. The sensor node distribution by Intel Labs is shown in Figure 4. The weather board sensor collects time stamp topology information as well as temperature, humidity, light and voltage values every 31 s. Due to node failure, there are several temperature values above 100 °C and below −30 °C. We, therefore, preprocessed the temperature data by taking the threshold of −5 °C and 45 °C by a priori knowledge.

In order to reduce the difference between the input data of the compressed model, and converge the algorithm more quickly, we mapped the original data to [0, 1] by max−min normalization, where max is the maximum value of the node sensing data, min is minimum value.
(14)$$x^{*}=\frac{x-min}{max-min}$$

After the preprocessing is completed, the node temperature database is converted to a data set. Each node collects temperature data as a column vector, and stores this data in the temperature.txt file. Due to node failure, some nodes only recorded a small amount of data. The average number of data by each node after preprocessing is 29,665. The maximum number of data in all nodes is 55,080, and the minimum number is 2507. We divide each node data into two parts: The training set and the test set. In order to ensure that each node has enough training data to train the model, we did not use common split standards such as 7:3 and 6:2:2, and instead used 9:1.

The more the number of samples in the training set, and the closer the empirical distribution is to the true distribution, the more accurate will the fitness of model distribution be. Increasing the number of training sets is an effective measure to prevent over-fitting. We provide following three methods: (1) Obtain data from the source; (2) estimate the data distribution parameters of the training set by statistical methods such as point estimation and interval estimation, and use this distribution to generate more data; (3) augment the training set by interpolation in the original training set, such as Kriging Interpolation and Natural Neighbor Interpolation. 

### 3.2. Compression and Reconstruction

We used the following performance criteria to evaluate the performance of our compression algorithm: (1) compression ratio (*CR*); (2) percentage RMS difference (*PRD*) [26]; (3) quality-score (*QS*) [26]. The definitions and formulas of these performance criteria are as follows:

(1) Compression Ratio (*CR*): It is defined as the compressed data length over the size of uncompressed data, as shown in Equation (15).
(15)$$CR=\frac{D_{or}}{D_{cp}}$$
where $D_{or}$ is the number of bytes of all original data. $D_{cp}\;$ is the number of bytes of all compressed data. The *CR* value is expected to be high for an effective compression algorithm.

(2) Percentage RMS Difference (*PRD*): It is a widely used performance measure that is used for calculating the quality of reconstructed data in the compression. The *PRD* value is expected to be as low as possible for a quality compression approach.
(16)PRD(%)=100×(∑i=0D−1(So(i)−Sr(i))2∑i=0D−1(So(i))2)12
where $S_{o}$ represents the original input data, $S_{r}$ represents the reconstructed data.

(3) Quality-Score (*QS*): Other important evaluation criteria for determining the effectiveness of compression algorithms is the *QS* value. *QS* is the ratio of *CR* to *PRD*. It represents the reconstructed data quality. The larger the *QS* value, the better the compression and reconstruction performance.
(17)$$QS=\frac{CR}{PRD}$$

The learning rate determines how far the weights move in the gradient direction in a mini-batch, which is usually set by the experimenter. If the learning rate is small, the training will become more reliable, but optimization will take a long time, because each step towards the minimum of the loss function is small. If the learning rate is big, the training may not converge and could also diverge. Optimization will cross the minimum value and cause loss function to become worse. There are many ways to set an initial value for the learning rate. A simple solution is to try a few different values to see which value will optimize the loss function without loss in training speed. In the pre-training phase, we refer to the parameters in Ref. [36], set the learning rate of 0.01/batch-size (the batch-size is 120). In the retraining phase, we start with a value of 0.1, then exponentially reduce the learning rate to 0.01, 0.001 and 0.0001. In order to find the optimal learning rate, we set different learning rates to train the model and record the summation of loss value when the model loss is no longer reduced [37]. Figure 5 shows the loss value of a model with different learning rates. We sought to find a point with the smallest value of model loss. In our experiment, we found that when the learning rate was between 0.0001 and 0.001, the model loss value was the smallest. We finally selected a learning rate of 0.0001.

We first use the training set of node 7 to pre-train the model without retraining, and test the efficiency of a different number of pre-training iterations to the compression performance of the model. Then we use the test set of node 7 to calculate the *PRD* value and the *QS* value of the model under different numbers of pre-training iterations. Similarly, we use the training set of node 7 to retrain the model with a fixed number of pre-training iterations. We use the test set of node 7 to calculate the *PRD* value and the *QS* value of the model under different numbers of retraining iterations. During the test, we added the *PRD* value and the *QS* value of each mini-batch of the test set, and then averaged the sum value as the final result. The number of mini-batches of the test set is 325. The results are shown in Figure 6.

Without retraining, the compression performance of the model will not increase and could even decrease with an increase in the number of pre-training iterations. When the number of pre-training iterations is 10 and the number of retraining iterations is 0, the compression performance of the model is optimal with the smallest *PRD* value and the largest *QS* value. Thus, we set the number of pre-training iterations of our model to 10. When the number of pre-training iterations is fixed, increasing the number of retraining iterations can significantly improve the compression performance of the model. When the number of retraining iterations reaches 200 and above, the compression performance of the model tends to be stable. At this time, increasing the number of retraining iterations does not significantly improve the compression performance of the model. Considering that the model needs to be calculated on the sensor node, increasing the number of retraining iterations will lead to an increase in calculation energy consumption. We finally selected the number of retraining iterations as 200.

In this experiment, we test the compression performance of the model under different *CRs*. This experiment data uses the data of node 7 with the number of pre-training iterations of 10, and the number of retraining iterations of 200. We then set *CR* as 10, 20, 40, and 120. We use the test set of node 7 to calculate the *PRD* value and the *QS* value of the model under different *CRs*. During the test, we summed up the *PRD* value and the *QS* value of each mini-batch of the test set, and then averaged the sum value as the final result. The number of mini-batches of the test set is 325. Figure 7 shows the compression performance of the model under different *CRs*.

Figure 7 illustrated that for a single node, with the increase of the *CR*, the compression performance of the model is not significantly increased or decreased, which shows that the model can imbibe the inherent properties of the data. These inherent properties are inherently weighted on the weight matrix and are independent of the dimension after compression. This is the difference between the deep compression method and the shallow compression method, since the reconstruction error of the shallow compression method increases with an increase in the *CR*. The result means that our algorithm can get a higher *CR* in the case of minimal reconstruction error. In our experiments, when *CR* was 10, the *PRD* value was the smallest. The reconstructed data value is closest to the original data value. Although increasing *CR* can significantly increase the *QS* value, the *PRD* value will also increase, which represents a difference between the reconstructed data, and the original data becomes larger. In this experiment, we explore the optimal compression performance of the model, and in the next experiment, we set the *CR* to 10 to explore the reconstruction performance of the model.

Figure 8 shows the reconstructed data and the original data of node 7 for our model, with the number of pre-training iterations being 10, the number of retraining iterations being 200, and *CR* of 10. We first use the model to compress the original data, and then use the model to reconstruct the compressed data. For all the data in the test set of node 7, we sum up the absolute value error between the original data and the reconstructed data, and then average the sum value as the final result. The average absolute value error obtained was 0.2815 °C, maximum value was 0.4602 °C, and minimum value was 0.0026 °C. Our model has been proven to have higher reconstruction accuracy, and the reconstructed data can correctly approximate the trend and value of the original data.

In the following experiments, we compare the performance of our algorithm with other compression algorithms. Performing CS algorithm on 40,000 data points for node 7. Since the length of stream data for CS algorithm cannot be too long, we divide these points into eight segments with segment length of 5000. We average the results of all segments as the final result of CS algorithm, and set the *CR* at 10. The results are shown in Table 1. At the same time, we test the performance of our algorithm on different data sets. The results are shown in Table 2.

### 3.3. Transfer Learning

In order to verify the generalization performance of the Stacked RBM-AE model, we use the data of node 7 to train the model, and then use the trained model to test the compression and reconstruction performance of all nodes. This process in deep learning is called transfer learning. At the same time, each node is trained separately to obtain the performance of the model of each node. We test the performance of the model after 10 pre-training iterations and 200 retraining iterations. We set the *CR* to 10 and the learning rate to 0.0001. We record the *PRD* value, *QS* value and reconstruction error of the model of each node. The *PRD* value and *QS* value of each node both are the average after summing up the value of *PRD* and *QS* of each mini-batch. The reconstruction error is the average after summing the absolute value error between the original data and the reconstructed data of each mini-batch. The red lines in Figure 9, Figure 10 and Figure 11 show the compression and reconstruction performance of the model of each node. The compression and reconstruction performance of each node on testing with parameters of node 7 are shown in blue lines in Figure 9, Figure 10 and Figure 11.

We can see from the red lines in Figure 9 and Figure 10 that the model has good compression performance for all nodes. For most nodes, the *PRD* value of the model is less than 20. There are some spikes in the red line in Figure 9, such as in node 18 and node 33. For this spike parts of the red line, we analyzed the model loss value during the training process, and find that the training loss value of the model is very small, reaching 3.7652, while the training loss value of most nodes are between 8 and 10. The phenomenon of low training error and high test error shows that the model has been over-fitted. Since we use a fixed number of training iterations in our experiments, the number of training iterations exceeds the optimal number of training iterations of the model for some nodes. By reducing the number of training iterations, the phenomenon of model over-fitting can be avoided effectively. The red line in Figure 11 shows that the error of data reconstruction is less than 1 for most nodes. Except those over-fitting nodes, the minimum and maximum error of data reconstruction can reach 0.2835 °C and 0.9303 °C for all nodes. Changing data cannot cause a significant drop in the compression and reconstruction performance of the model. The blue line shows that the model trained with the data of node 7 is not very suitable for most other nodes. Compared to the red lines, the *PRD* value and reconstruction error value are increased. For all nodes, the biggest difference of the *PRD* value between red line and blue line can reach 15.37, and the smallest difference is 0.16. The minimum and maximum error of the digital difference between the data reconstruction of red line and blue line is 0.0176 °C and 0.5289 °C, respectively. For nodes located near node 7, such as node 6, node 8 and node 9, the reconstruction error varies little and the performance of the model does not decrease significantly by using model parameters that are not corresponding to the nodes themselves. The digital difference of reconstruction error of node 6, node 8 and node 9 is 0.0875 °C, 0.1127 °C and 0.0720 °C, respectively. This means that for some areas of the WSNs with strong spatial correlation, when a node in the area is trained, all nodes in the area can share the trained model parameters. These results prove that our algorithm has a good transfer learning ability. The error reduction speed during the model training helps judge whether the training is completed or not. When the training error value drops slowly and approaches a stable value, it means that the training is complete. In our experiments, when the training of the model was completed, the number of training iterations was mostly less than 200. This can further reduce the storage and communication energy consumption of node model parameters. The transfer learning ability can be applied to the design of clustering routing protocol in WSNs.

### 3.4. Energy Optimization

For the use of the Stacked RBM-AE model in WSNs, we present a solution. The server first uses the node data to train the model and sends the trained model parameters to the node (only the parameters of the encoder need to be sent, because the node does not need to decode the data). After the node receives the trained model parameters, the node constructs and initializes the encoder by using the parameters. When the node collects the sensing data, the encoder is used to encode and compress the data. The compressed data is formed into a data packet. Then the node sends the data packet to the server. The server uses the decoder to decode and reconstruct the sensing data of the data packet. For some areas of the WSNs with strong spatial correlation, the server only needs to train one time of one node in the area to get the model parameters, and all nodes in the area share the parameters. Figure 12 shows the process.

In our experiments, we use the model to compress the sensing data; when the number of data is 120, we set *CR* to 10. If the type of original sensing data is float, the original data size is 120 × 4 = 480 (byte), and the packet size after compression is 12 × 4 = 48 (byte). The energy of data transmission can be reduced by 90% for each node. Using the Stacked RBM-AE model to compress a data packet needs 18,737 floating point calculations (including multiplication and addition). We train the model by using NVIDIA GeForce GTX 1080 Ti and Tensorflow. Training the model take 181.54 ms when the number of model training iterations is 200 and the number of sensing data is 120. Compressing this sensing data takes 0.23 ms. We also test the floating-point calculation speed of STM32F103. When the main frequency of the STM32F103 is 150 M, STM32F103 will take 42 instruction cycles (0.22 ms) to execute a floating-point calculation. In theory, STM32F103 will take 4122 ms to calculate the encoder floating point calculation of the model, where the main frequency is 150 M. We utilize the parameter pruning method to further reduce storage energy consumption and communication energy consumption. Figure 13 shows the energy optimization process. Figure 14 shows the parameter pruning process.

We judge the importance of the parameters in the model according to the absolute value of the parameters [38], and then calculate the number of pruning and prune threshold, according to the prune ratio. The parameters in the model whose absolute value is less than the prune threshold are removed from the network, and then we retrain the model to restore the performance of the model.

Algorithm 3 shows the procedure of the pruning parameters. We first sort the trained model weights according to the absolute value, and then multiply the prune rate and the number of weight parameters to get the number of parameters that we need to prune. After sorting is completed, we expand the weights into a 1-D vector. The numerical value is found in the vector as the prune threshold according to the number of prune parameters. The weights are compared with the prune threshold value. When the absolute value of the weights is less than the prune threshold, set the value to 0. We get a simplified model that initialized the model parameters by using the pruned weights. In order to restore accuracy, this simplified model requires retraining to fine-tune weights, because we prune a portion of the original model parameters [39]. We then iterate the pruning and retraining processes until the model performance returns to the original performance.


**Algorithm 3: Pruning Parameters. ABS(x) indicate the absolute value for x. Sort(x) indicate sorting by the numerical value of x.**
**Input:** The model weights **W**, the prune rate **α**, the number of prune iterations *iter***While**
*i < iter*
**do**  Sort(ABS(**W**))  Get m which is the number of elements in **W**  Get the number of **W** that need pruning n=α×m  Get prune threshold ***thr***  **if W** < ***thr***
**then**    **W** = **0**  **end**  Update parameters W according to Algorithm 2  *i + +***end**

The network of the Stacked RBM-AE model has eight layers, and each layer of the network is a standard RBM. We use ***L_i_*** to represent the i layer standard RBM in the model. The distribution of the Stacked RBM-AE model parameters is shown in the Table 3. As we can see from Table 3, most of the model parameters are located in ***L*_1_**, ***L*****_2_**, ***L*_7_** and ***L*_8_**. When we prune weights, we only need to prune the weights of the ***L*_1_**–***L*_4_**, because the weights of the ***L*_5_**–***L*_8_** is the transposition of the ***L*_1_**–***L*_4_** weights.

In order to explore the efficiency of pruning the weights of the different layers on the performance of the model, we separately prune different rates of the weights of each layer. We use the data of node 7 to train and test the model. During the test, we record the average value of *PRD*, *QS* and reconstruction error of each mini-batch of the test set. We use $W_{i}$ to represent the weights of the $i_{th}$ layer. Figure 15 shows the compression performance of the model under different prune rates of each layer.

When the prune rate is below 20%, pruning the weight respectively of any layer will not lead to the decline of the performance of the model. This means that there are redundant data in the weights of our model, which cannot affect the performance of the model. On the whole, the compression performance decreases with the increasing the prune rate. Removing too many parameters from the model will affect the data learning and fitting ability of the model. If we seek to increase the prune rate while maintaining the performance of the model, we need to choose an appropriate prune rate according to the model compression performance change.

## 4. Conclusions

In this paper, we propose a Stacked RBM-AE model to compress WSNs data by using RBM and AE. In order to improve compression performance, we design a model parameter adjustment strategy, which includes two parts: pre-training and retraining. We test the efficiency of the number of iterations of pre-training and retraining on the performance of the model by experiment studies. We also offer a solution to use the model for WSNs and discuss the computational efficiency of the model. Considering the calculation and communication energy consumption, we use the method of prune parameters to further optimize the energy consumption of our algorithm. Our experimental results show that our algorithm has better transfer learning ability, and has better reconstruction accuracy than the traditional algorithms under the same *CR*. The data communication energy consumption can be reduced by 90%.

The sensor node usually be equipped with multiple sensors to collect different environmental monitoring data. This method can be extended to perform joint compression and reconstruction of multi-stream data. Since Gibbs Sampling and k-step divergence are used to estimate the probability distribution of the reconstructed data in this method, the theoretical system error is inevitably introduced. Exploring how to reduce the systematic error of the model is the focus of our next research work. VAE, GAN or other mixed deep learning models can also be used in future research work, which is expected to reduce the systematic error caused by the model assumptions and further reduce the energy consumption for WSNs. We will also look at how to use deep neural networks for WSNs.

## Figures and Tables

**Figure 1 sensors-18-04273-f001:**
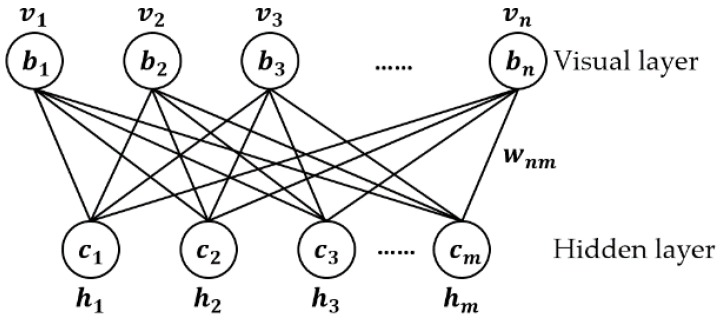
Standard RBM.

**Figure 2 sensors-18-04273-f002:**
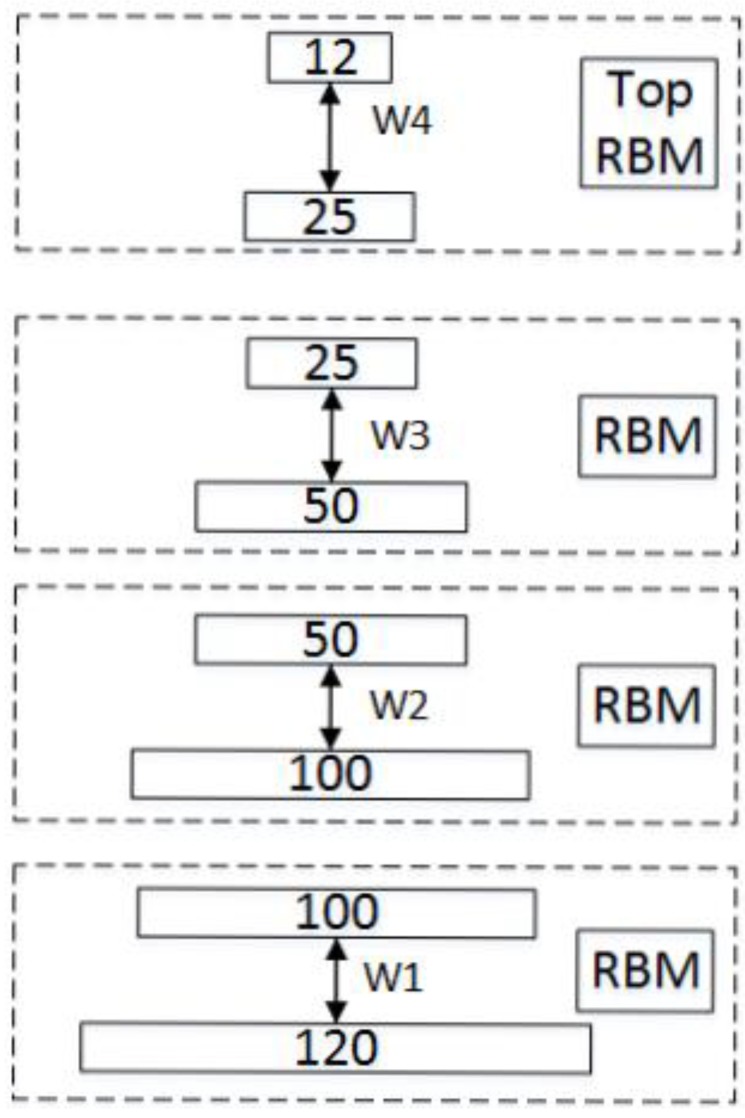
Stacked RBM.

**Figure 3 sensors-18-04273-f003:**
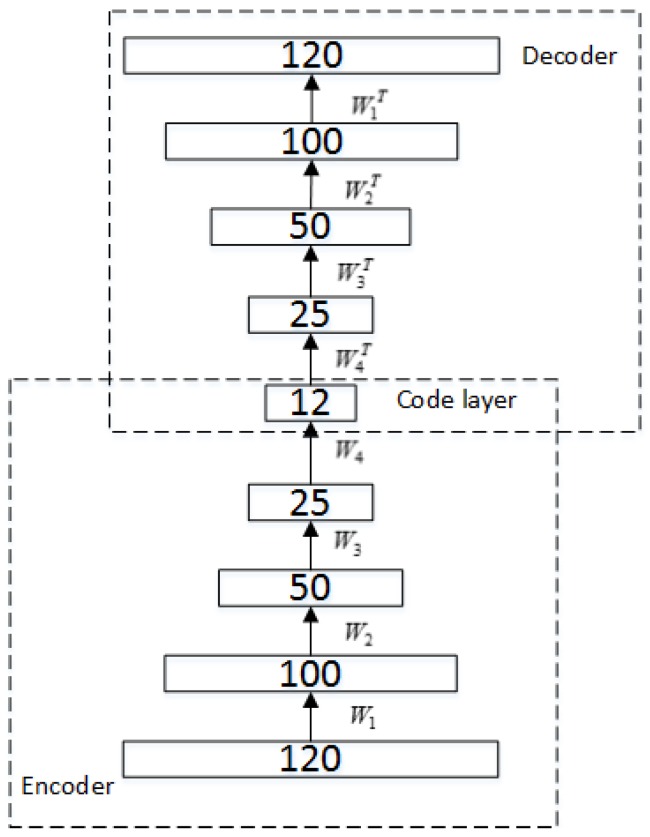
Stacked RBM-AE.

**Figure 4 sensors-18-04273-f004:**
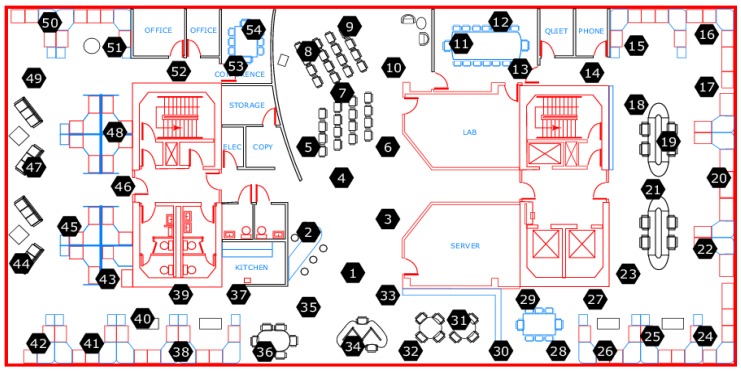
Sensor node distribution in Intel Lab.

**Figure 5 sensors-18-04273-f005:**
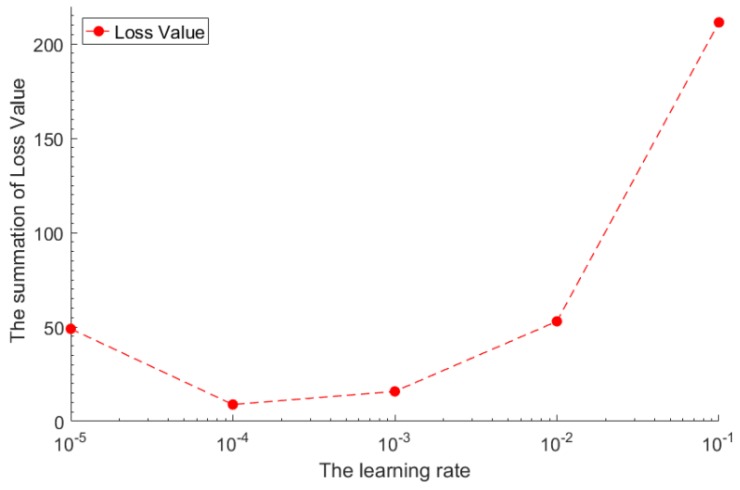
The loss value of model with different learning rates.

**Figure 6 sensors-18-04273-f006:**
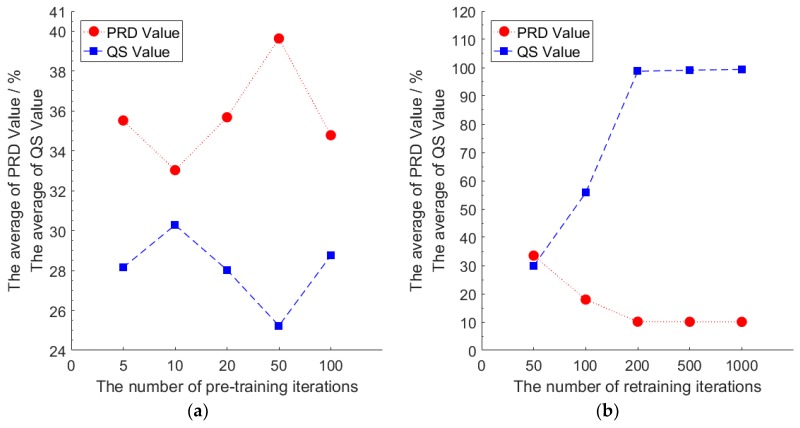
(**a**) Compression performance under different number of pre-training iterations without retraining; (**b**) compression performance under different retraining iterations with the number of pre-training iterations as 10.

**Figure 7 sensors-18-04273-f007:**
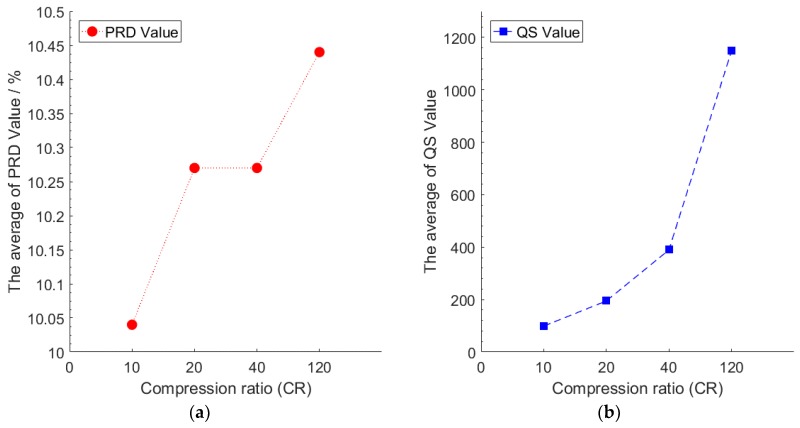
(**a**) The *PRD* value under different *CRs*, (**b**) The *QS* value under different *CRs.*

**Figure 8 sensors-18-04273-f008:**
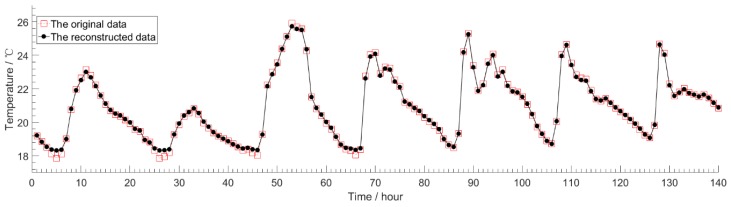
The reconstructed data and the original data of node 7 under *CR* is 10.

**Figure 9 sensors-18-04273-f009:**
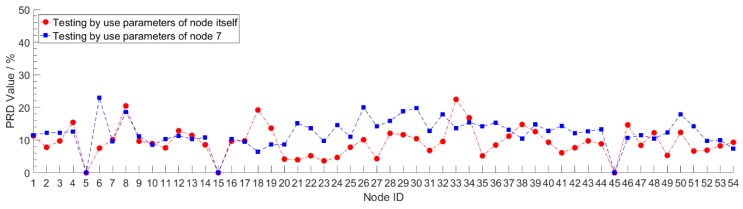
The *PRD* value of each node on testing with different model parameters.

**Figure 10 sensors-18-04273-f010:**
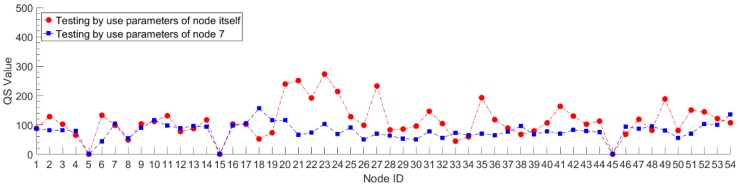
The *QS* value of each node on testing with different model parameters.

**Figure 11 sensors-18-04273-f011:**
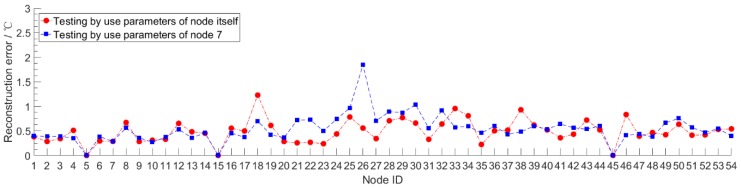
The reconstruction error of each node on testing with different model parameters.

**Figure 12 sensors-18-04273-f012:**
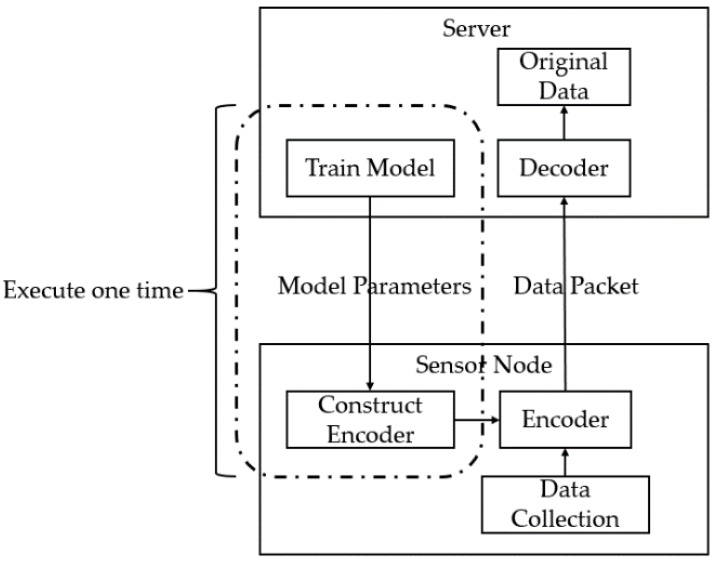
The process of the model solution.

**Figure 13 sensors-18-04273-f013:**
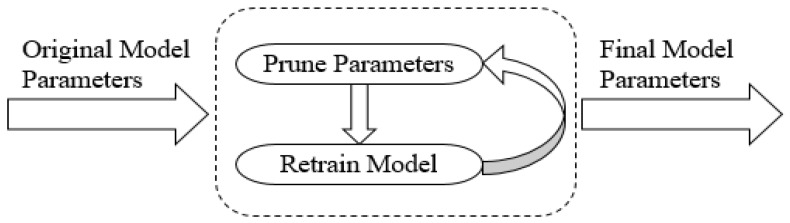
Energy optimization.

**Figure 14 sensors-18-04273-f014:**
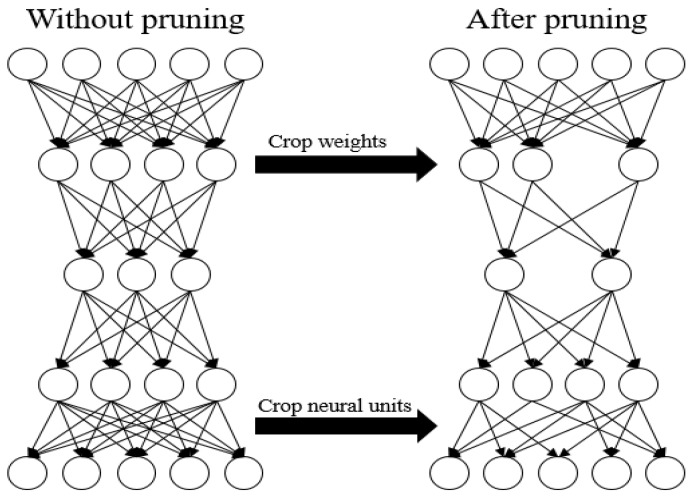
Parameter pruning.

**Figure 15 sensors-18-04273-f015:**
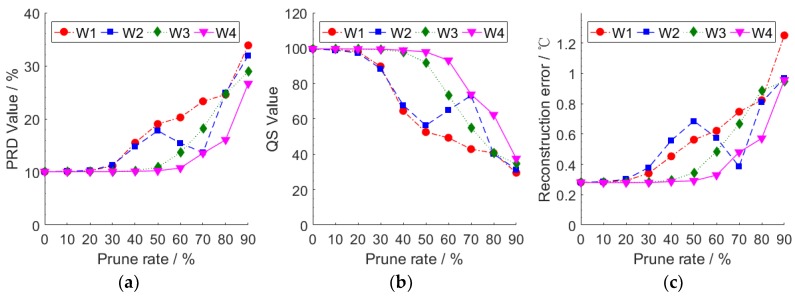
(**a**) The *PRD* value under different prune rates of each layer; (**b**) the *QS* value under different prune rates of each layer; (**c**) the reconstruction error under different prune rates of each layer.

**Table 1 sensors-18-04273-t001:** Compression performance comparison.

Algorithm	*PRD* (%)	*QS*	Reconstruction Data Error (°C)
CS	38.40	26.04	1.4143
Standard RBM	33.03	30.28	1.0423
Our algorithm	10.04	99.60	0.2815

**Table 2 sensors-18-04273-t002:** Model performance on other datasets.

Dataset	*PRD* (%)	*QS*	Reconstruction Data Error
Argo (temperature)	11.10	90.09	0.8434 (°C)
ZebraNet (location/UTM format)	9.82	101.83	259.26
CRAWDAD (speed)	8.53	117.23	6.2056 (km/h)
Intel Lab (humidity)	10.90	91.74	3.8383 (%RH)

**Table 3 sensors-18-04273-t003:** The distribution of the Stacked RBM-AE model parameters.

Layers	Parameters Number	Storage (Byte)	Proportion (%)
$L_{1}$	12,100	48,400	32.20
$L_{2}$	5050	20,200	13.44
$L_{3}$	1275	5100	3.39
$L_{4}$	312	1248	0.83
$L_{5}$	325	1300	0.86
$L_{6}$	1300	5200	3.50
$L_{7}$	5100	20,400	13.57
$L_{8}$	12,120	48,480	32.25

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
