# Peer review of "Data Compression Based on Stacked RBM-AE Model for Wireless Sensor Networks"

_sensors, 2018, doi:10.3390/s18124273_

Reviewer 1 Report

This manuscript proposed a stack RBM-AE model to achieve data compression in wireless sensor networks. The proposed model is composed of a code layer and a decode layer, which can effectively reduce the energy consumption of model storage and calculation. Finally, experimental data were used to evaluate the performance of the proposed data compression model. Overall, the topic of this study is interesting and the structure is well organised. I suggest the paper can be published in Sensors if the authors well answer the following comments.

1. In introduction, a comprehensive literature review on existing data compression algorithms for WSN is required.

2. The authors should provide which platform is used to implement the proposed algorithm. Besides, the analysis about calculation efficiency is necessary.

3. More data compression algorithms should be compressed to proposed one based on same data set to demonstrate its superiority.

4. More future research should be included in conclusion.

5. There are several obvious typos that affect the quality of the manuscript. Please revise them.

Author Response

Dear reviewer,

First of all, I apologize for the troubles that my poor English writing ability has brought you. And thank you very much for your recognition and your valuable comments. This time, I modify my manuscript according to your suggestion. At the same time, my classmates help me correct the many serious English spelling, typos and grammar errors in manuscripts.

Point 1: In introduction, a comprehensive literature review on existing data compression algorithms for WSN is required.

Response 1: We extend the introduction section, and conduct a comprehensive literature review on existing data compression algorithms for WSNs. At the same time, we investigate and discuss the background and technology of the proposed algorithm. The new introduction is shown on lines 28-114 in the new manuscript.

Point 2: The authors should provide which platform is used to implement the proposed algorithm. Besides, the analysis about calculation efficiency is necessary.

Response 2: In the new manuscript, we explain which platform that we use to implement and test the proposed algorithm. In addition, we test and analyse the calculation time and calculation amount of the proposed algorithm. (on lines 407-419 in the new manuscript)

Point 3: More data compression algorithms should be compressed to proposed one based on same data set to demonstrate its superiority.

Response 3: We add other data compression algorithm experiments based on the same data set and the results are shown in Table 1. (on line 343 in the new manuscript) We do a standard RBM data compression experiment. The results can demonstrate the superiority of the proposed algorithm. At the same time, we test the performance of the proposed algorithm on different data sets. The results are shown in Table 2. (on line 344 in the new manuscript)

Point 4: More future research should be included in conclusion.

Response 4: In the conclusion section, we add a description of more future research that we will do, including the proposed algorithm improvements, and other algorithmic applications for WSNs. (on lines 465-483 in the new manuscript)

Point 5: There are several obvious typos that affect the quality of the manuscript. Please revise them.

Response 5: We revise some English spelling, typos and grammar errors in the new manuscript. But there may be some missing errors we have not found.

Reviewer 2 Report

Abstract should focus on novel work proposed by the authors and include quantitative results.  Concisely focus on novelties proposed, with sufficient details on conditions, assumptions, specific method features.  Salient quantitative results and compact comparison with the state of the art should conclude the abstract.

The authors should present and explain the rationale behind using RBMs instead of generative adversarial networks or variational autoencoders, as most ML practitioners currently do.

Proposed method and experimental results should be compared with existing state of the art to underline its novelty and progress.

Also, experimental results should include statistical parameters, like min/max/average/STD.  For instance, clarify how you calculated the temperatures reported on lines 237–238 and their variability for other data sets to support the robustness and stability of the proposed method.

Related work should be significantly expanded to include most relevant state of the art works that cover the many techniques addressed by the method proposed by the authors.  E.g., [1,2] are rather old and do not cover well techniques for significantly reducing network traffic by processing sensing data on the WSN nodes as much as possible (see, for instance, [a,b] below).  [c–f] below are some examples of recent works addressing similar issues.  The authors should freely include any reference they deem relevant to proposed method in order to thoroughly compare them with the advantages and limitations of the proposed method.

A reference for "compression sensing" should be added (top of right column, page 1).

Algorithm 1 should be moved and explained towards begin of Section 2.2 to help the reader follow the related equations.  Similarly, Algorithm 2 should be moved and discussed before the related equations.

Design decisions should be motivated and alternatives comparatively discussed and quantified.  For instance, "appropriate length" (line 191), "120 temperature data" and "1 hour" (line 193), authors' "great relevance" belief on line 194, the 9:1 split between training and testing (commonly 70-30 or 60-20-20), "suitable number" on line 224–225, how was tested the effect (line 225), learning rate of 0.0002 on lines 226–227, pre-training time of 10 (measurement unit?)  on line 233, retraining time 200 (measurement unit?)  on line 236.

Option 2) on line 200 should be reformulated more clearly.  Same for similarities with image processing at point 3), line 202.  Also discuss why not augment sensor data using interpolations.

Size of the file on line 207 should be replaced by a scientific measurement, since file size can vary with the magnitude of the numbers, representation format, accuracy, etc.

The rationale behind using CR = 10 on line 222 and later on should be carefully explained.

Discuss the meaning of the plots shown in Figures 8–10, and the spikes in red lines in and the corresponding high difference from blue lines.  Also motivate and quantify your conclusion that compression performance does not decrease significantly (line 277), and how much different is sensor data with which you tested this assumption.

Quantitatively compare the performance of proposed method with state of the art compression methods (using RBM or other techniques).

Improve clarity of the explanations on lines 286–290, perhaps with the support of a diagram.

Also clarify how can the nodes infer when they can reuse the pre-trained models.

Explain how pruning was implemented and the rationale behind main implementation decisions.  Also provide motivation and discussion of other decisions, such as to use eight layers (line 299).

Figures 5, 8–10, 12 should be referenced and explicitly and adequately discussed in text.

Captions of figures, tables, algorithms are not informative.  They should be self-explanatory.  The authors should not expect the reader to be well acquainted with the text to understand the meaning of tables and figures.  Specific symbols used in figures, tables or algorithms should be briefly explained in the captions.

Figure 4 lacks a legend and text should be larger to be readable.  Figures 5, 7-10, 13 should include measurement units for values on the axes and have the symbols used explained in captions.

Symbols used in the manuscript should be collected and explained in a table before their use, to increase the readability of the article.

Text on line 151 should be emphasized like the one on line 106.

The manuscript should be carefully proofread, if possible by a native English speaker, to correct the many serious English spelling, typos and grammar errors.

---

[a] Xiang, L., Luo, J. and Vasilakos, A., 2011, June.  Compressed data aggregation for energy efficient wireless sensor networks.  In Sensor, mesh and ad hoc communications and networks (SECON), 2011 8th annual IEEE communications society conference on (pp.  46-54).  IEEE.

[b] Lazarescu, M.T., 2013.  Design of a WSN platform for long-term environmental monitoring for IoT applications.  IEEE Journal on emerging and selected topics in circuits and systems, 3(1), pp.45-54.

[c] Tramel, E.W., Manoel, A., Caltagirone, F., Gabrié, M. and Krzakala, F., 2016, September.  Inferring sparsity: Compressed sensing using generalized restricted Boltzmann machines.  In Information Theory Workshop (ITW), 2016 IEEE (pp. 265-269).  IEEE.

[d] Norouzi, M., Ranjbar, M. and Mori, G., 2009, June.  Stacks of convolutional restricted boltzmann machines for shift-invariant feature learning.  In Computer Vision and Pattern Recognition, 2009.  CVPR 2009.  IEEE Conference on (pp. 2735-2742).  IEEE.

[e] Hinton, G.E. and Salakhutdinov, R.R., 2006.  Reducing the dimensionality of data with neural networks.  science, 313(5786), pp.504-507.

[f] Hinton, G.E. and Salakhutdinov, R.R., 2012.  A better way to pretrain deep boltzmann machines.  In Advances in Neural Information Processing Systems (pp. 2447-2455).

Author Response

Dear reviewer,

First of all, I apologize for the troubles that my poor English writing ability has brought you. And thank you very much for your recognition and your valuable comments. This time, I modify my manuscript according to your suggestion. At the same time, my classmates help me correct the many serious English spelling, typos and grammar errors in manuscripts.

Point 1: Abstract should focus on novel work proposed by the authors and include quantitative results. Concisely focus on novelties proposed, with sufficient details on conditions, assumptions, specific method features.  Salient quantitative results and compact comparison with the state of the art should conclude the abstract.

Response 1: We modify the abstract according to your suggestion. We change some descriptions of the abstract and briefly describe the details of the proposed algorithm. At the same time, we give our quantitative experimental results. In the end, we summarize the abstract. (on lines 9-24 in the new manuscript)

Point 2: The authors should present and explain the rationale behind using RBMs instead of generative adversarial networks or variational autoencoders, as most ML practitioners currently do.

Response 2: Compared with VAE and GAN, RBM has the simplest network structure and the minimum number of parameters. Correspondingly, RBM has a small computational energy consumption and is more suitable for use in WSNs with limited energy. Currently, we mainly explore how to use RBM to compress the sensing data. In our next research, we will also explore how to use VAE, GAN and other deep learning models to compress the sensing data.

Point 3: Proposed method and experimental results should be compared with existing state of the art to underline its novelty and progress.

Response 3: We add other data compression algorithm experiments based on the same data set and the results are shown in Table 1. (on line 343 in the new manuscript) We do a standard RBM data compression experiment. The results can demonstrate novelty and progress of the proposed algorithm. At the same time, we test the performance of the proposed algorithm on different data sets. The results are shown in Table 2. (on line 344 in the new manuscript)

Point 4: Also, experimental results should include statistical parameters, like min/max/average/STD.  For instance, clarify how you calculated the temperatures reported on lines 237–238 and their variability for other data sets to support the robustness and stability of the proposed method..

Response 4: In the new manuscript, we refine our specific experimental steps and we enrich our experimental content. We give the calculation details of our experimental results for all experiments, and add the statistical parameters of the experimental results. At the same time, we discuss the results of our experiments. We modify the description about the results on lines 237-238 in the original manuscript. The new description is on lines 330-333 in the new manuscript.

Point 5: Related work should be significantly expanded to include most relevant state of the art works that cover the many techniques addressed by the method proposed by the authors.  E.g., [1,2] are rather old and do not cover well techniques for significantly reducing network traffic by processing sensing data on the WSN nodes as much as possible (see, for instance, [a,b] below).  [c–f] below are some examples of recent works addressing similar issues.  The authors should freely include any reference they deem relevant to proposed method in order to thoroughly compare them with the advantages and limitations of the proposed method..

Response 5: We extend the introduction section, and conduct a comprehensive literature review on existing data compression algorithms for WSNs. At the same time, we investigate and discuss the background and technology of our algorithm. The new introduction is shown on lines 28-114 in the new manuscript.

Point 6: A reference for "compression sensing" should be added (top of right column, page 1).

Response 6: Sorry, I don't know the meaning of “top of right column, page 1”. We add the related literature of compression sensing to the reference.

Point 7: Algorithm 1 should be moved and explained towards begin of Section 2.2 to help the reader follow the related equations.  Similarly, Algorithm 2 should be moved and discussed before the related equations.

Response 7: We modify the order of the algorithms and the related equations, and we discuss the algorithms before the related equations. At the same time, we enrich the specific details of the algorithms. (on lines 145-150, 187-200 in the new manuscript)

Point 8: Design decisions should be motivated and alternatives comparatively discussed and quantified.  For instance, "appropriate length" (line 191), "120 temperature data" and "1 hour" (line 193), authors' "great relevance" belief on line 194, the 9:1 split between training and testing (commonly 70-30 or 60-20-20), "suitable number" on line 224–225, how was tested the effect (line 225), learning rate of 0.0002 on lines 226–227, pre-training time of 10 (measurement unit?)  on line 233, retraining time 200 (measurement unit?)  on line 236.

Response 8: We modify the description of the corresponding location in the original manuscript (on line 191,193 and 194 in the original manuscript). In the new manuscript, we give the reason of why we set the node data split ratio is 9:1. (on lines 231-235 in the new manuscript) We describe in detail how we test the performance of the proposed algorithm and discusse the detailed results of our experiments. (on lines 278-302 in the new manuscript) We describe detailedly how we obtain the parameters of the proposed algorithm, including the learning rate, the number of pre-training iterations and the number of retraining iterations. (on lines 261-302 in the new manuscript) We also add measurement units to all experimental data.

Point 9: Option 2) on line 200 should be reformulated more clearly.  Same for similarities with image processing at point 3), line 202.  Also discuss why not augment sensor data using interpolations.

Response 9: We modify the corresponding description. We introduce a simple method to the proposed point and some examples. We add the point and some examples of using interpolation methods to enhance data. We remove the point of image processing because we mainly discuss streaming data, and the correlation between image and stream data is low. (on lines 239-242 in the new manuscript)

Point 10: Size of the file on line 207 should be replaced by a scientific measurement, since file size can vary with the magnitude of the numbers, representation format, accuracy, etc.

Response 10: In the new manuscript, we use the number of bytes of the data to replace the size description in the original manuscript. (on lines 247-251 in the new manuscript)

Point 11: The rationale behind using CR = 10 on line 222 and later on should be carefully explained.

Response 11: We modify the order of the experiments to make the experimental sequence look more reasonable. We test the performance of the proposed algorithm under different CRs. The results show that the proposed algorithm has the best reconstruction performance when CR is 10. (on lines 303-326 in the new manuscript)

Point 12: Discuss the meaning of the plots shown in Figures 8–10, and the spikes in red lines in and the corresponding high difference from blue lines.  Also motivate and quantify your conclusion that compression performance does not decrease significantly (line 277), and how much different is sensor data with which you tested this assumption.

Response 12: In the new manuscript, we redefine the experimental protocol. We discuss the results of the new experiment results and give the specific digital results. At the same time, we discuss and explore the phenomena in the result graphs, and give the reasons and solutions for this phenomenon. We also give the method of how to judge the model training is completed. (on lines 346-393 in the new manuscript)

Point 13: Quantitatively compare the performance of proposed method with state of the art compression methods (using RBM or other techniques).

Response 13: We did a standard RBM data compression experiment. (on line 343 in the new manuscript) At the same time, we test the performance of the proposed algorithm on different data sets. (on line 344 in the new manuscript)

Point 14: Improve clarity of the explanations on lines 286–290, perhaps with the support of a diagram.

Response 14: We modify the description of the solution of how to use the proposed model in WSNs. We describe detailedly each step of the solution. At the same time, we draw a figure to describe the specific process of the program. (on lines 395-406 in the new manuscript)

Point 15: Also clarify how can the nodes infer when they can reuse the pre-trained models.

Response 15: The node does not need to infer when they can reuse the pre-trained models. The parameters of node is given by the server. We give the method of how to judge the model training is completed. (on lines 387-390 in the new manuscript)

Point 16: Explain how pruning was implemented and the rationale behind main implementation decisions.  Also provide motivation and discussion of other decisions, such as to use eight layers (line 299).

Response 16: We describe detailedly the specific steps of the pruning algorithm and the principles of pruning. At the same time, we discuss and give the results of the performance of the proposed model under different prune rates of each layer. (on lines 424-464 in the new manuscript)

Point 17: Figures 5, 8–10, 12 should be referenced and explicitly and adequately discussed in text.

Response 17: In the new manuscript, we discuss adequately all the experiment result figures and give the calculation process of the results in this figures.

Point 18: Captions of figures, tables, algorithms are not informative.  They should be self-explanatory.  The authors should not expect the reader to be well acquainted with the text to understand the meaning of tables and figures.  Specific symbols used in figures, tables or algorithms should be briefly explained in the captions.

Response 18: In the new manuscript, we explain briefly the meaning of the specific symbols. The symbols used in tables and algorithms are explained in corresponding tables and algorithms. The symbols used in figures are shown in corresponding figures.

Point 19: Figure 4 lacks a legend and text should be larger to be readable.  Figures 5, 7-10, 13 should include measurement units for values on the axes and have the symbols used explained in captions.

Response 19: In the new manuscript, we provide the legend of Figure 4 and change the size of Figure 4. The measurement units for values on the axes and symbols used in this figures are shown in corresponding figures.

Point 20: Symbols used in the manuscript should be collected and explained in a table before their use, to increase the readability of the article.

Response 20: In the new manuscript, we collect all the symbols used in our manuscript, but the number of symbols is too large and cause the table size become too large. So we give up the collection measures. We explain the symbols in corresponding used location.

Point 21: Text on line 151 should be emphasized like the one on line 106.

Response 21: We modify the text format of the corresponding location. (on line 182 in the new manuscript)

Point 22: The manuscript should be carefully proofread, if possible by a native English speaker, to correct the many serious English spelling, typos and grammar errors.

Response 22: We revise some English spelling, typos and grammar errors in the new manuscript. But there may be some missing errors we have not found.

Round  2

Reviewer 1 Report

The authors well answered the comments, so I suggest that current version can be published in Sensors.

Author Response

Dear reviewer, Thank you very much for your help on my manuscript.

Reviewer 2 Report

Response 2 (quoted below) is likely interesting for readers and should be included in the manuscript:

Response 2: Compared with VAE and GAN, RBM has the simplest network structure and the minimum number of parameters.  Correspondingly, RBM has a small computational energy consumption and is more suitable for use in WSNs with limited energy.  Currently, we mainly explore how to use RBM to compress the sensing data.  In our next research, we will also explore how to use VAE, GAN and other deep learning models to compress the sensing data.

Figure 7 (a and b) and (17) are not coherent.  According to (17), quality score (QS) should decrease for higher compression ratios (CR) in Figure 7b, since Figure 7a shows that percentage RMS difference (PRD) increases with the increase of CR.

Figures 5–7 should include statistical parameters (min, max, STD, average) and the number of experiments used to generate them should be reported.

Author Response

Dear reviewer,

Thank you very much for your help on my manuscript.

Point 1: Response 2 (quoted below) is likely interesting for readers and should be included in the manuscript:

«Response 2: Compared with VAE and GAN, RBM has the simplest network structure and the minimum number of parameters.  Correspondingly, RBM has a small computational energy consumption and is more suitable for use in WSNs with limited energy.  Currently, we mainly explore how to use RBM to compress the sensing data.  In our next research, we will also explore how to use VAE, GAN and other deep learning models to compress the sensing data.»

Response 1: We add this content in the introduction part of the new manuscript. (on lines 93-99 in the new manuscript)

Point 2: Figure 7 (a and b) and (17) are not coherent.  According to (17), quality score (QS) should decrease for higher compression ratios (CR) in Figure 7b, since Figure 7a shows that percentage RMS difference (PRD) increases with the increase of CR.

Response 2: The unit of measurement for PRD is %. QS is the ratio of CR to PRD. In our manuscript, we use the real value data of PRD to calculate QS. Figure 7(a) shows the PRD value of the proposed model when CR is 10, 20, 40 and 120. The PRD value are 10.04%, 10.27%, 10.28% and 10.44%, respectively. Correspondingly, the QS value are 99.60 (10/0.1004), 194.74 (20/0.1027), 389.11 (40/0.1028) and 1149.43 (120/0.1044). The PRD value increases slightly (only 0.4%) with the increase of CR. So Figure 7 (a and b) looks not coherent.

Point 3: Figures 5–7 should include statistical parameters (min, max, STD, average) and the number of experiments used to generate them should be reported.

Response 3: For Figures 5–7, we add the corresponding statistical parameters. In Figure 5, we record the summation of loss value as the final result when the model loss is no longer reduced. (on lines 275-277 in the new manuscript) In Figures 6–7, we sum the PRD value and the QS value of each mini-batch of the test set, and then average the sum value as the final result. The number of mini-batchs of the test set is 325. (on lines 288-290, 312-314 in the new manuscript)